**Data Availability Statement:** We have added our data to OSF (DOI 10.17605/OSF.IO/ZFYEX – https://osf.io/zfyex/).

**Funding:** This work was supported by an NSERC Discovery Grant to EPZ and an NSERC Post-

# 1894 revisited: Cross-education of skilled muscular control in women and the importance of representation

Gregory E. P. Pearcey [1,2,3,4], Lauren A. Smith[2,3,4], Yao Sun[2,3,4,5], E. Paul Zehr[2,3,4,6]*

**1** Department of Physiology and Physical Medicine and Rehabilitation, Feinberg School of Medicine, Northwestern University, Chicago, IL, United States of America, **2** Rehabilitation Neuroscience Laboratory, University of Victoria, Victoria, British Columbia, Canada, **3** Human Discovery Science, International Collaboration on Repair Discoveries (ICORD), Vancouver, British Columbia, Canada, **4** Centre for Biomedical Research, University of Victoria, Victoria, British Columbia, Canada, **5** Department of Physical Therapy, Faculty of Rehabilitation Medicine, University of Alberta, Edmonton, Alberta, Canada, **6** Division of Medical Sciences, University of Victoria, Victoria, British Columbia, Canada

* pzehr@uvic.ca

## Abstract

In 1894 foundational work showed that training one limb for "muscular power" (i.e. strength) or "muscular control" (i.e. skill) improves performance in both limbs. Despite that the original data were exclusively from two female participants ("Miss Smith" and "Miss Brown"), in the decades that followed, such "cross-education" training interventions have focused predominantly on improving strength in men. Here, in a female cohort, we revisit that early research to underscore that training a task that requires precise movements in a timely fashion (i.e. "muscular control") on one side of the body is transferred to the contralateral untrained limb. With unilateral practice, women reduced time to completion and the number of errors committed during the commercially available game of Operation® Iron Man 2 with both limbs. Modest reductions in bilateral Hoffmann (H-) reflex excitability evoked in the wrist flexors suggest that alterations in the spinal cord circuitry may be related to improvements in performance of a fine motor task. These findings provide a long overdue follow-up to the efforts of Miss Theodate L. Smith from more than 125 years ago, highlight the need to focus on female participants, and advocate more study of cross-education of skilled tasks.

## Introduction

In the year 1894, Edward Wheeler Scripture, Miss Theodate L. Smith and Miss Emily Brown [1] published "On the education of muscular control and power." This study had only two participants, both women, who were also co-authors. Miss Brown (as identified in the paper) trained for 'muscular power' (strength training) and Miss Smith for 'muscular control' (skill training). Their training was performed with one arm for 9 and 10 days, respectively, and both their trained arm and the contralateral untrained arm improved strength and accuracy by 40% and 25% with 'muscular power' training and 'muscular control' training, respectively. Their work was the first to formally identify 'cross-education', which is the bilateral improvement in

Graduate Scholarship to GEP. The Operation®
games were purchased by Zanshin Consulting Inc.

**Competing interests:** The authors have declared
that no competing interests exist.

performance with unilateral training. Since that time, an entire field of research [2] has studied
various aspects of cross-education (sometimes referred to as inter-limb/inter-manual transfer
or the cross-transfer effect). Early experiments focused on acute (within session) learning with
one hand, and showed that learning novel tasks transferred quite well to the contralateral
untrained limb. Many studies have since characterized the cross-education of 'muscular
power' described by Scripture, Smith and Brown, however only one study has closely resem-
bled the cross-education of 'muscular control'. In this study, Schulze et al. [3] examined the
effects of unilateral training of a timed pegboard task and found bilateral improvements in
time to completion after training. Although indirectly assessed (i.e. the timer kept running
when a peg was dropped), there was no direct assessment of accuracy that coincided with
improved time to completion. Therefore, this task did not directly assess what Scripture,
Smith, and Brown originally coined as the cross-education of 'muscular control'.

Underlying mechanisms of cross-education have been identified within cortical and
subcortical regions, and within the spinal cord [4–6]. For example, reductions in interhemi-
spheric inhibition from the ipsi- to contralateral motor cortex is well-correlated with the
intermanual transfer of a serial reaction time task after ~30 minutes of motor sequence train-
ing [7]. This transcallosal pathway seems to dominate whichever model is brought forward as
the leading framework for the neural processes underlying cross-education, whether "cross-
activation" or "bilateral access". Although not mutually exclusive, the two models commend
modest subtleties that differentiate themselves from one another. The "cross-activation"
hypothesis suggests that both cortices are activated to a certain extent during unilateral behav-
iours, whereas the "bilateral access" hypothesis suggests that motor plans that are formed
through practice of one limb may be accessed in the future for the performance with the con-
tralateral limb [5]. In both cases, the transcallosal pathway appears essential to the cross-educa-
tion of performance, as evidenced by the increases in cortical activation in the ipsilateral
sensorimotor cortex with unilateral handgrip training [8]. Below the cortical and sub-cortical
regions, cross-education of strength causes alterations in reciprocal inhibition and Hoffmann
(H-) reflex excitability that accompany improvements in strength suggesting that cross-educa-
tion also causes plasticity of spinal origin [9]. Whether this occurs in response to cross-educa-
tion of skill is less clear, even though it is well known that H-reflex excitability within the
trained limb is reduced immediately following (within 10 minutes) acquisition of a novel fine
motor skill [10].

The clinical relevance of cross-education of strength has recently been highlighted, such
that strength training the less affected limb can facilitate strength gains and functional
improvements in the untrained, more-affected limb in chronic stroke participants with accom-
panying neurophysiological changes at the spinal and supraspinal level [11, 12]. Although
cross-education of strength at the ankle joint has shown functional improvements in walking
[11], functional improvements in hand function (i.e. clinical tests of hand function) did not
accompany the observed improvements in strength as a results of cross-education in the upper
limb [12]. To target functional improvements in upper limb function, we suggest that the
cross-education of skill in a rehabilitation setting could be an alternative to the cross-education
of strength.

Cross-education spares muscle strength and size when a limb is immobilized and the oppo-
site limb is trained [13]. Restoring symmetry of strength after musculoskeletal or neurological
impairment through cross-education of strength has therefore gained substantial traction
lately [14], given the vast literature available to support its efficacy (for reviews see [14, 15]).
Cross-education of skill has received less attention in the clinical realm, possibly because of the
inadequate support it has received in the literature since the majority of studies have focused
on more acute transfer effects (within session to < 2 weeks) (for examples see [7, 16–23]). In

only a couple studies, more long-term training effects were examined, but were limited to the fifth digit adduction/abduction visuomotor tracking [24] and visuomotor tracking of simple elbow flexion/extension [25]. Although these tasks are great experimental paradigms to assess changes in fine motor control, they are not very functional tasks. Therefore, we set out to closely replicate the experiment of Scripture, Smith, and Brown [1] and examine whether 5 weeks of unilateral training for highly functional 'muscular control' results in bilateral improvements in functional motor performance. True to the original work and in direct contrast to the vast majority of cross-education studies specifically and strength training studies generally, we included a women-only sample of participants.

To replicate in modern day the approach taken in 1894 that used an electrified drillboard, we had participants play the game of Operation® Iron Man 2 (furthermore referred to simply as "the game") and used a combination of the time to completion and number of contacts (i.e. errors) as an index of task performance. This game requires people to use tweezers to reach into small holes and pull out plastic objects without touching the metallic edges because a contact will cause a harsh buzzer to sound. Therefore, it is a highly functional task that requires precise movements in a timely fashion (i.e. high muscular control) and closely resembles the electrified needle and peg board with bell used in 1894. We hypothesized that, similar to acute transfer found in other skilled tasks, bilateral improvements in task performance would manifest from unilateral training. Furthermore, we hypothesized that improved 'muscular control' would be accompanied by reductions in spinal reflex excitability, which would provide novel insights about the neural mechanisms contributing to the cross-education of skill.

## Materials and methods

### Participants

Nine neurologically intact young women (aged 22–24) were recruited for this study. Eight of the participants were right handed, while the other was left-handed as assessed by the Edinburgh handedness inventory [26]. The sample size was determined based on prior related work in the laboratory using similar designs and outcome measures, and were sufficient to achieve significant cross-education effects of strength training with moderate effect sizes [9, 12, 27]. Participants signed a written consent form that adhered to the protocol approved by the University of Victoria Human Research Ethics Committee.

### Experimental design

The experimental timeline is in Fig 1B. This study used a multiple baseline within subject repeated-measures design, where participants completed three baseline measures and one post-test measure [12, 27, 28]. Multiple baseline tests were used to enable participants to act as their own controls. These baseline measures were obtained roughly a week apart and in order to maintain consistency, measures were recorded in the same order and environment across sessions. The post-test was completed within a week after the last training session.

Bilateral measures of muscular control and strength, muscle activation, and reflex excitability were assessed during the pre- and post-test. Muscular control was quantified by examining errors and time to completion for one round of with each hand during pre- and post-testing. Muscular strength was quantified by measuring handgrip and pinch grip strength. Muscle activation was measured using electromyography (EMG) of the flexor carpi radialis (FCR), extensor carpi radialis (ECR), biceps (BB), and triceps brachii (TB) during muscular strength tests. Reflex excitability was assessed using H-reflex recruitment curves during a low-level FCR contraction (10% MVC).

## Unilateral training of muscular control

Participants completed five weeks of unilateral functional training using the game for three sessions/week (twice at home and once in the lab). Each session included five rounds of the game with their dominant hand. The goal of the game was to take all 11 pieces out of Iron Man without touching the metal edges of the holes with the tweezers. An instructed order of piece removal was not specified, and participants could remove in any order of their choosing. If the tweezers touched the metal edge, an alarm sounded and an error was counted. In keeping with the early literature, participants were not explicitly instructed how to hold the tweezers (5.4cm in length with a 7mm aperture), but the most common way was to hold the tweezers between the tips of the thumb and index finger with support from the middle finger. Objects ranged in size with graspable regions ranging from 1-4mm, while the holes ranged in size with the narrowest region being 16mm (see Fig 1A for shapes of objects and holes). This

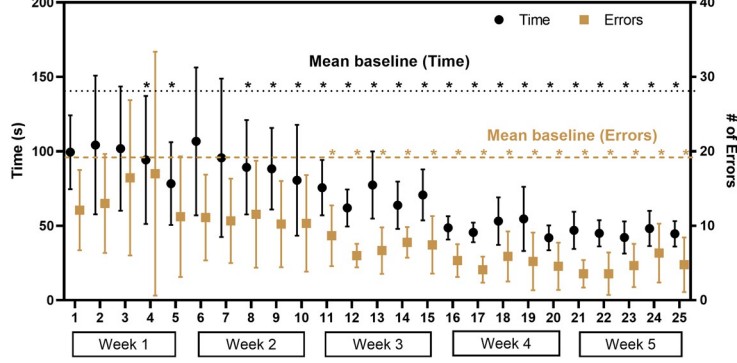

**Fig 1.** A) A photo of the Operation® Iron Man 2 game (Hasbro, Canada) used in experiments, courtesy of G Pearcey. Objects that were removed are highlighted in green and the tweezers used by participants are highlighted in magenta. B) Miss Smith's results from 1894. The percent of trials with errors are plotted on the primary y-axis for the untrained (yellow) and trained (black) hand. Lines indicate the number of trials performed per day and are plotted against the secondary y-axis. C) The experimental timeline. D) The group (n = 9) mean (± 95% CI) time to completion (black; primary y-axis) and number of errors committed (gold; secondary y-axis) over the training protocol. Each time point represents a single trial from the weekly training session that was performed in the laboratory. Asterisks indicate significant differences from the pre-test mean.

game was selected as the closest commercially available approximation of the original apparatus from 1894, which was an electrified drill board into which a needle was inserted. Contact with the circumference of the drill hole closed the circuit and rang a bell. Each participant used the same game for their pre and post training measures as well as their training in the lab. However, while training at home participants had their own game to practice on but this game was an identical model to the game that they were tested on. Errors, identified by an electrified buzzer, and completion time were only recorded in lab training sessions, however feedback about the number of errors committed was not provided to participants.

## Assessing muscular control

The primary outcome measure was completing the game in as little time as possible, while committing as few errors as possible, rather than simply completing a given number of trials without time restriction. Hence, we assessed both time to completion and the number of times the buzzer sounded (i.e. number of errors). During each of the three pre tests, participants played the game one time with each hand, first with the dominant hand (trained) and then with the non-dominant (untrained). Participants did not play the game with their untrained hand again until the post test. Therefore, we did not track the improvements in the untrained hand to minimize any effect from repeated tests. The game of Operation® Iron Man 2 is similar to the electrified pegboard that was initially used by Edward Scripture to examine the cross-education of skill, however the added time constraint probably added another layer of difficulty.

## Assessing muscular strength

Maximal voluntary force during handgrip and pinch grip were evaluated using a hand grip dynamometer and MICROFET2 force evaluation device, respectively. Participants completed three attempts of maximal voluntary contractions (MVCs) in each hand for hand and pinch grip tests. MVCs were held for three seconds and separated by two minutes of rest. Strength assessment was recorded in a seated position with the non-tested handed placed in their lap.

## Hoffmann (H) reflexes as a proxy of spinal cord plasticity

Spinal cord excitability was estimated by evoking H-reflexes in the FCR. The role of the FCR during gameplay is somewhat unclear, since its primarily involved in wrist flexion and radial deviation, but it does act at the wrist and is almost certainly involved in synergies for wrist stabilization and subtle wrist movements that contribute to task performance. Additionally, other training tasks show "spillover" to other muscles that could be detected with the approach we took. Therefore, due to the methodological convenience and experience of our lab, we evoked H-reflexes of the FCR as a proxy of spinal reflex excitability in distal arm muscles by delivering 1 ms square wave pulses to the median nerve just proximal to the medial epicondyle with bipolar surface electrodes (Thought Technology Ltd., Montreal, QC, Canada) using a Digitimer (Medtel, NSW, Australia) constant current stimulator (model DS7A). Current delivered for each stimulus was measured with a non-contact milliammeter (mA-2000, Bell Technologies, Orlando, FL, USA). H-reflexes were recorded in a seated position with the non-tested hand resting in their lap. Their tested arm was placed in customized brace that restricted movement and maintained joint angles. Each arm was fixed with the: 1) shoulder in 30 and 15 degrees abduction and flexion, respectively, 2) elbow at 110 degrees, and 3) wrist pronated with the fingers open and strapped to a wooden fixture. These measurements were taken with a manual goniometer by the same investigator for consistency. Participants were asked to maintain a low-level wrist flexion contraction (~10% of maximum), while they received visual feedback

from a computer screen. Feedback consisted of a 100 ms moving average of the rectified EMG. H-reflexes were induced and recorded following procedures that have been described elsewhere [29–31]. To examine input-output properties of the H-reflex pathway, M-H recruitment curves were measured over a range of intensities with 40 stimuli delivered pseudorandomly between 1 and 3 s. Stimulus intensity was increased and decreased incrementally (ranged from 0.1 to 1 mA per increment) based on inter-individual differences in the excitability of the reflex pathway. Careful attention was taken to ensure that supramaximal M-wave amplitudes were achieved by increasing larger increments once the H-reflex amplitude started to decrease in size. Peak to peak amplitudes of the H and M waves were calculated using custom written software (Matlab, Nantick, MA) and data was then imported into custom written LabView software where it was fit with a sigmoid function [32]. We normalized the stimulation current to that required to evoke 50% of $M_{max}$ and amplitudes of M-waves and H-reflexes to $M_{max}$. We then combined the amplitude vs current arrays for all responses in the three pre tests prior to performing a sigmoid fit. The normalized and combined recruitment curve was then compared with the sigmoid fit of the post recruitment curve. Detailed descriptions of all recruitment curve variables can be found in Klimstra and Zehr [32]. Briefly, the *current at threshold* was the relative stimulation current required to evoke the smallest H-reflex, *current at 50% $H_{max}$* was the relative stimulation current required to evoke an H-reflex 50% of maximum amplitude, *current at $H_{max}$* was the relative stimulation current required to evoke the maximum H-reflex. Using relative current from the pre recruitment curves, we derived comparative values at the same relative current at post. These included the H-reflex size at the current required to evoke the smallest H-reflex from pre, H-reflex size at the current required to evoke 50% of the maximal H-reflex from pre, and H-reflex size at the current required to evoke the maximal H-reflex from pre. The sigmoid fit was used to obtain these values with the procedures outlined in Klimstra and Zehr [32]. In a subset of the participants (n = 3), current intensity values were compromised from at least one of the H-reflex recruitment curves throughout the timeline. Therefore, only Hmax/Mmax ratios are reported for the entire study sample (n = 9), whereas recruitment curve variables are reported for a subset of participants that had current intensity values for both limbs and all time points (n = 6).

## Electromyography

Prior to placement of surface electrodes, the skin was prepared with isopropyl alcohol swabs. Electrodes were placed bilaterally over the mid-muscle bellies of the FCR, ECR, BB and TB with an inter electrode distance of 2cm and common reference electrodes were placed on the medial epicondyles. Electrode placement was recorded at the initial baseline test to ensure each electrode was placed in the correct orientation in each subsequent test. During H-reflex recordings, FCR amplification was set to ×2000 and filter settings were adjusted to 10-1000Hz, whereas all other muscles were amplified ×5000 and band pass filtered 100–300 Hz (GRASS P511, AstroMed). Outputs were sent to the A/D interface (National Instruments Corp. TX, USA) and converted to a digital signal. EMG was sampled at 1000 Hz using custom built software (LabVIEW, National Instruments, TX, USA).

## Statistical analysis

Statistical procedures were performed using GraphPad Prism (GraphPad Software, San Diego, CA). Separate 2-way (Limb × Time) repeated measures (RM) ANOVAs were used to determine whether there were main or interaction effects of time or limb on the dependent variables of time to completion, number of errors, hand grip strength, pinch grip strength, and maximal H-reflex amplitudes. Assumptions of sphericity and normality were confirmed using

Mauchly's and Kolmogorov-Smirnov tests, respectively. If significant effects of time were identified, Bonferroni's multiple comparisons tests were used. For the trained limb, we used a 1-way RM ANOVA to determine if there was a main effect for time on time to completion or number of errors throughout the sessions in the lab. If significant effects of time were identified, Dunnett's multiple comparison test was used to determine differences of all time points from pre. For recruitment curve measures, the variables were obtained from a sigmoid fit [32] and separate 2-way (Limb × Time) repeated measures (RM) ANOVAs were used to determine whether there were main or interaction effects of time or limb on each of the variables. In all cases, statistical significance was set at $p \leq 0.05$. Results are reported as means ± SD in text (95% CI in figures). In addition to group statistics, the multiple baseline design allowed us to quantify the number of participants who showed significant improvement over the duration of the experiment by creating a 95% confidence interval from the three baseline tests. If a post-test score was below the lower limit of the baseline confidence interval, it was deemed a significant improvement in time to completion or number of errors.

## Results

### Timeline of training effects for the trained limb

Fig 1D shows that the time to complete and number of errors during one game of Operation® Iron Man 2 reduced quickly. Separate one way RM ANOVAs revealed that there were significant effects of time for both the time to complete ($F_{(25, 200)} = 5.596$, $p < 0.0001$, $\eta^2 = 0.41$) and number of errors ($F_{(25, 200)} = 3.481$, $p < 0.0001$, $\eta^2 = 0.30$) during one game of Operation® Iron Man 2. Times to completion for all games played in the lab, except the first three games of week one, and both the first and second games of week 2, were significantly less than pre (see Fig 1D–black asterisks). Errors were reduced during all games played in the lab sessions of weeks 3, 4 and 5 (see Fig 1D–gold asterisks).

### On muscular control

**Time to completion.** The overall time to complete one game of Operation® Iron Man 2 did not differ between pre tests, but was significantly reduced for both the trained and untrained limb after the 5 week training program (see Fig 2A). The RM ANOVA revealed a significant effect of time ($F_{(3, 48)} = 17.56$, $p < 0.0001$, $\eta^2 = 0.36$), but no effect of limb ($F_{(1, 16)} = 2.628$, $p = 0.125$, $\eta^2 = 0.02$) or an interaction ($F_{(3, 48)} = 1.776$, $p = 0.164$, $\eta^2 = 0.036$). Post test time to completion was significantly less than pre 1 ($p < 0.0001$), 2 ($p < 0.0001$) and 3 ($p = 0.0014$), however there were no significant differences between pre test values, or between the trained and untrained limb. Compared to baseline, a total of 8/9 and 6/9 participants

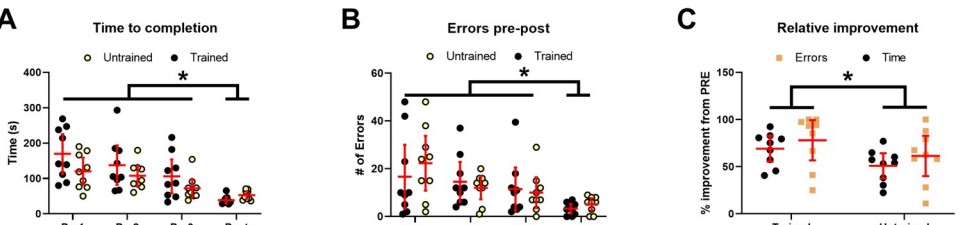

**Fig 2.** Individual and group (n = 9) mean (± 95% CI) values are shown with individual points and red bars, respectively, for A) time to completion and B) number of errors, and C) relative improvement for both the trained and untrained limbs. Asterisks indicate significant differences from all pre values for A and B, and a significant difference between the trained and untrained limbs in C.

improved their time to completion over the duration of the training program for the trained and untrained limb, respectively.

**Number of errors.** The total number of errors committed by participants during one game of Operation® Iron Man 2 did not differ between pre tests, but was significantly reduced for both the trained and untrained limb after the 5 week training program (see Fig 2B). The RM ANOVA revealed a significant effect of time (F(3, 48) = 9.35, p < 0.0001, $\eta^2$ = 0.23), but no effect of limb (F(1, 16) = 0.058, p = 0.813, $\eta^2$ = 0.001) or an interaction (F(3, 48) = 0.803, p = 0.499, $\eta^2$ = 0.002). The number of errors in the post test was significantly less than pre 1 (p < 0.0001), 2 (p = 0.0004) and 3 (p = 0.002), however there were no significant differences between pre test values, or between the trained and untrained limb. Compared to baseline, a total of 5/9 and 5/9 participants significantly reduced the number of errors committed over the duration of the training program for the trained and untrained limb, respectively. It is important to note, however, that any participant who committed 0 errors on any of their pre tests or who had a confidence interval that extended into negative values, a significant improvement was not possible. Indeed, this was the case for 2 participants' trained limb and 3 participants' untrained limbs. Thus, conclusions based on these individual participant results must be made with caution as they likely underestimate the number of participants with significant improvements in performance.

**Relative improvement.** The relative improvement of both time to completion and number of errors is plotted in Fig 2C. Although there is a general trend for performance to improve, the mean of all three pre tests was used to calculate the relative pre-post improvement. A 2-way (limb × measure) RM ANOVA revealed a significant effect of limb (F(1, 16) = 7.507, p = 0.0145, $\eta^2$ = 0.13) but no effect of measure (F(1, 16) = 1.191, p = 0.291, $\eta^2$ = 0.04) or an interaction effect (F(1, 16) = 0.0148, p = 0.905, $\eta^2$ = 0.002). The trained limb showed ~15% greater improvement than the untrained limb across measures.

## On muscular power

Although hand grip (F(1, 16) = 11.11, p = 0.0011, $\eta^2$ = 0.889) strength was higher for the trained (pre = 26.1 ± 6.0 kg, post = 25.41 ± 4.94 kg) compared to untrained limb (pre = 23.0 ± 6.35 kg, post = 22.89 ± 5.62 kg), there were no significant effects of time or interaction effects. No significant effects were revealed for pinch grip strength (trained pre = 3.65 ± 1.25 kg, trained post = 3.77 ± 1.07 kg, untrained pre = 3.42 ± 1.13 kg, trained post = 3.26 ± 1.18 kg).

## On excitability of H-reflexes

The overall H-reflex excitability was modestly reduced in the trained and untrained limbs after the 5 week training program (see Fig 3). In the subset of participants with reliable stimulation current intensity data (n = 6), the RM ANOVA revealed a significant effect of time, but no effects of limb or interactions for maximal H-reflex amplitude relative to maximal M-wave amplitude (Time main effect: F(1, 10) = 6.18, p < 0.0323, $\eta^2$ = 0.12), and threshold current required to evoke the H-reflex (Time main effect: F(1, 10) = 6.42, p = 0.029, $\eta^2$ = 0.25). Across limbs, maximal H-reflex amplitudes decreased by 9.35% $M_{max}$, and threshold current required to evoke the smallest possible H-reflex increased by 18% of relative current. A similar trend was revealed in the $H_{max}/M_{max}$ ratio for the entire group (n = 9) and can be appreciated from a single participant in Fig 4. Group mean $H_{max}/M_{max}$ ratio was reduced by 9.5 ± 8% (pre $H_{max}/M_{max}$ ratio = 41 ± 10.8%, post $H_{max}/M_{max}$ ratio = 36.6 ± 13.8%, d = 0.51) in the trained limb and 3.2 ± 7% (pre $H_{max}/M_{max}$ ratio = 51 ± 20.6%, post $H_{max}/M_{max}$ ratio = 49.4 ± 12.2%, d = 0.23) in the untrained limb from pre- to post-training, however, despite the moderate

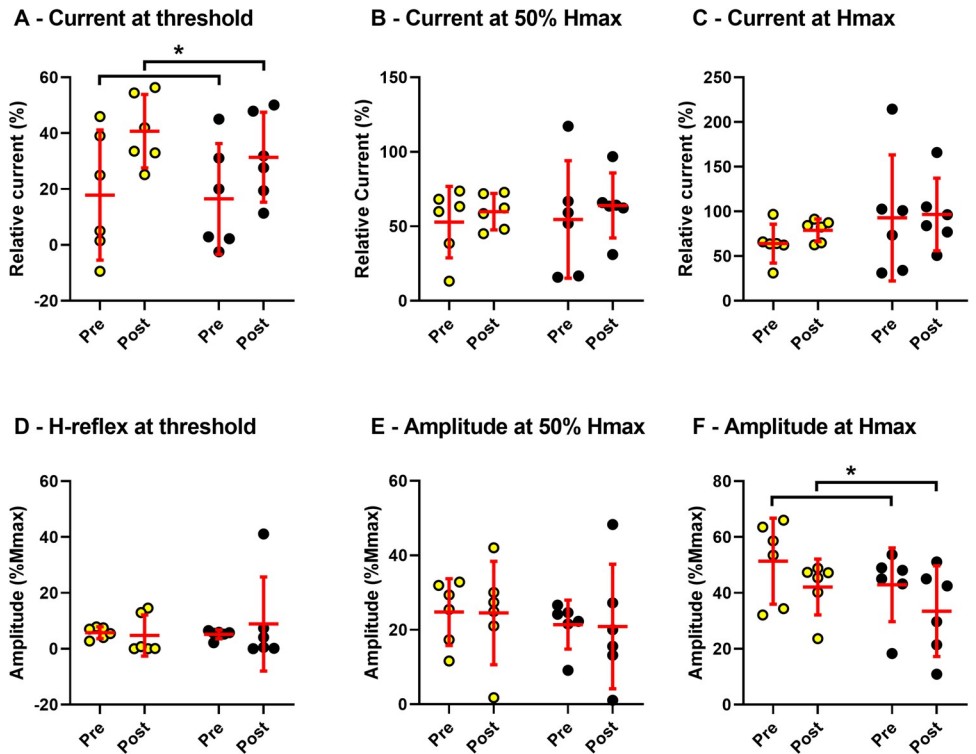

**Fig 3. Individual and group (n = 6) mean (± 95% CI) values are shown with individual points and red bars.** In all panels, the untrained limb is represented with yellow fill, whereas the trained limb is represented with black fill. A-C: the current required to evoke: H-reflex threshold; 50% Hmax; and, Hmax. D-F: the amplitudes of H-reflexes at the current from pre to evoke: threshold; 50% Hmax; and, Hmax. Detailed descriptions of all variables can be found in Klimstra and Zehr [2008]. Asterisks indicate significant main effects of time.

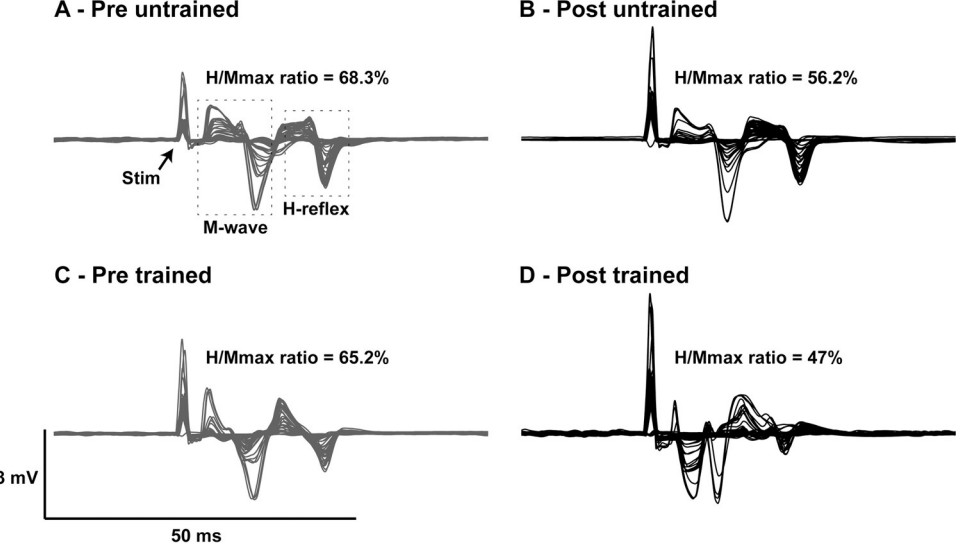

**Fig 4.** Individual EMG records for each stimulation pulse (n = 40) are shown for a participant's pre (left; black) and post (right; grey), and untrained (top) and trained (bottom) FCR M-wave and H-reflex recruitment curves. All traces are overlaid and $H_{max}/M_{max}$ ratios are indicated for the trial as a percent of $M_{max}$.

Cohen's d values, no statistical differences were observed. $M_{max}$ amplitudes ranged from 2.792 to 13.271 mV on the untrained side and from 2.871 to 14.746 mV on the trained side but no significant effects were revealed for limb or time.

## Discussion

The most important findings from this study were 1) a ~70% improvement in both time to completion and number of errors committed with the trained limb; 2) a ~55% improvement in the time to completion and number of errors committed with the untrained contralateral limb; and, 3) a modest reduction in bilateral H-reflex excitability from pre- to post-training. Collectively, these findings suggest that training for 'muscular control' with one limb causes bilateral improvements in performance and that reductions in spinal reflex excitability may contribute to the improved task performance. Moreover, and most importantly, our work highlights the critical importance for additional studies focusing on female participants both out of need, necessity, and equity, and out of respect for and deference to the foundational contributions to the field of cross-education from Emily Brown and Theodate Smith.

### Improvements with the trained limb

It is quite surprising how quickly and the overall extent to which participants improved their performance (see Fig 1D). Improvements were observed in the first week of training for time to completion and within the second week for the number of errors committed. Observed improvements manifest more quickly than those reported for strength training [27], but align well with studies examining the learning of a novel skilled motor task. In such studies, improved performance of the trained limb is noted within a session that can be less than 30 minutes in duration [20]. Interestingly, the relative extent of improvement (i.e. % change) is > 2 times as large as the improvements in strength during similar length training interventions [33]. These findings demonstrate that practice of a task that involves 'muscular control' is effective at improving fine motor control, evidenced by improvements in both the time to completion and the number of errors committed.

### Improvements with the untrained limb

Training the dominant limb for 'muscular control' resulted in performance improvements on the contralateral untrained side as well. We cannot comment on the time course of the cross-education of 'muscular control' because our experimental timeline would not allow, however, we can point out the relative improvement in task performance in the contralateral untrained limb compared to the trained side. In particular, it is fascinating to see that both the trained and untrained sides showed immense improvement in both time to completion and number of errors committed (see Fig 2C). Of note however, is that the untrained side relative improvement was greater than 50% on average, which is substantially greater than the improvements that have been shown in the literature for the cross-education of 'muscular power'. A meta-analysis by Green and Gabriel [33] showed that the group average relative improvement in strength from unilateral training ranges from 15–35% and 12–29% on the trained and untrained sides, respectively. This results in a cross-body transfer (i.e. relative improvement on the untrained side compared to the trained side) that ranges from 48–80%. In our study, the cross-body transfer was on the higher end, but still within this range (72.6% for time to completion and 91.3% for number of errors committed), suggesting that the cross-education of muscular control is at least as effective as the cross-education of 'muscular power'. This corroborates Scripture and colleagues [1] report that showed similar cross-body transfer for 'muscular power' (i.e. 57%) as they did for 'muscular control' (i.e. 55%). Together, these findings

demonstrate that cross-education of 'muscular control' is highly effective, as evidenced by the large relative improvement with the untrained limb compared to the trained limb and the substantial cross-body transfer of the training effect.

As suggested by the "bilateral access" and "cross-activation" hypotheses [5], it is likely that transcallosal pathways are crucial for the cross-education of both muscular control and power. In both cases, it is highly likely that motor plans become available for the contralateral limb for subsequent performance (i.e. bilateral access), and ipsilateral pathways are activated during unilateral practice (i.e. cross-activation). As such, it is no surprise that similar adaptations are often observed along the corticospinal pathway between strength and skill training [34]. Where the modalities differ is in the specific circuits mediating such neural adaptations. When training for improvements in strength, increased activation of the agonist and reduced activation of antagonist muscles is vital to improved force output about a joint, which can be attributed to changes throughout the neuraxis [35, 36]. When training for improvements in fine motor control, precise control and coordination of the joint(s) are most important. Compared to self-paced strength training, unilateral strength training to a metronome and visuomotor (i.e. skill) training result in bilateral adaptations in corticospinal excitability and inhibitory cortical circuits (i.e. short-latency intra-cortical inhibition). Such adaptations in the ipsilateral cortex underlie the mechanisms proposed within the cross-activation hypothesis, suggesting that training for skill facilitates cortical adaptations associated with the cross-activation hypothesis [37]. As such, greater contributions of the cross-activation hypothesis associated with training that involves a skill could explain the greater extent of adaptations in the contralateral limb when training for muscular control (this study), compared with previous reports of the cross-education of strength.

### Reduced spinal reflex excitability—An additional potential site for an underlying mechanism?

Previous work has shown that training for 'muscular control' causes reductions in H-reflex excitability [10]. Here, we have provided preliminary evidence for the reduction in bilateral H-reflex excitability of the upper limb as a result of training for 'muscular control' using the Operation® Iron Man 2 game. Since a subset of participants was used to examine H-reflex excitability, it is crucial to acknowledge to the limitations of these conclusions. For instance, we cannot conclude that the reduction in H-reflex excitability was functionally relevant and contributed to improved game performance, or whether it was a passive adaptation that occurred as a result of upstream modifications in descending neural drive. For instance, it is possible that corticospinal tracts could increase inhibition of the Ia terminals through increased presynaptic inhibition [38], which would help with fine motor control. Our data are sparse and further work is required to elucidate the neural mechanisms responsible for the cross-education of 'muscular control'.

### Important considerations

It is important to emphasize a few limitations that should be considered before drawing conclusions from this brief research report. For example, although there were no significant changes across baseline measures, visual inspection of our data may suggest a trend towards improvement in these sessions. Future studies may consider multiple post-test measures as a way to evaluate this. Further, we did not directly assess the kinematics and muscle activity during gameplay, thus limiting our ability to assess what control strategy was altered to improve performance. When participants committed an error a buzzer sounded, and there is potential that the buzzer caused some startle, especially early in gameplay, which caused time loss during

the startle response. However, with repeated errors throughout training, it is possible that the participants became accustomed to the buzzer sound and therefore lost less time during each error due to this reduced startle. Although no obvious startle responses were observed in initial gameplay by the investigators, we cannot exclude this possibility. This habituation to the startle response make help explain the counterintuitive trend for reduced time to complete the game with the non-dominant hand, compared to the dominant hand (i.e. see Fig 2A).

The small sample size of participants in this study from which reliable H-reflex recruitment curves could be obtained limits the conclusions that can be drawn from the alterations observed in spinal reflex excitability, however, it still suggests that bilateral inhibition of spinal reflexes may play a role in the acquisition of fine motor skills resulting from unilateral training. Related to this, the small sample size limits our ability to assess correlations between alterations in reflex excitability and adaptations in muscular control. As such, it is unknown whether the change in reflex excitability was related to training improvements, or whether it occurred en passé. The role of the muscle we used to assess spinal reflex excitability in the successful completion of the Operation® Iron Man 2 game is not well characterized. Sampling from, for example first dorsal interosseous, thenar muscles, or flexor digitorum superficialis would be directly relevant to the pinch grip tweezer type task and fine movements of the fingers during gameplay. Therefore, any future study should try and explore excitability changes in intrinsic hand or finger flexor/extensor muscles during tasks that train muscular control.

Our sample of subjects was young (22–24) and the application to older individuals may be perceived as limited. It should be noted, however, that cross-education of strength has been observed in our lab well into their 7th decade [11, 12]. Most importantly, however, this work highlights the value of single participant case studies and illustrates some false assumptions underlying what is actually control versus control group in scientific papers. We also note with some irony the incredible contributions two young women (i.e. Ms. Smith and Ms. Brown) made to what became an enormous research enterprise (i.e. cross-education) spawning hundreds of papers but that studies using women as participants have been largely neglected in the century that followed. An exception to this trend has come from the work of Jon Farthing, who examined the application of cross-education of strength in women on multiple occasions [39–41], and two other studies that compared cross-education of strength between young and older women [42, 43]. Investigations of the effects of cross-education of skill in women, however, requires further work.

## Conclusions

Although the work of Scripture, Smith and Brown [1] had only one woman for each of the conditions cross-education of 'muscular control' and cross-education of 'muscular power', their findings established an entire field of research and remain influential more than a century later. Here we corroborate and extend their findings to show, in a cohort of women, that unilateral training of a task that requires precise movements in a timely fashion (i.e. high muscular control) causes bilateral improvements in task performance. We also show that training for such 'muscular control' causes bilateral reductions in H-reflex excitability that may or may not contribute to improvements in fine motor control. These findings suggest that, like the cross-education of strength, cross-education of skill should be further examined for efficacy in the rehabilitation of musculoskeletal and neurological impairments. Lastly, we hope that our study focusing exclusively on women helps encourage more studies in female participants. The legacy of Miss Smith and Miss Brown, along with their colleague Edward Scripture, must be continuously acknowledged, preserved, and enhanced.

## Author Contributions

**Conceptualization:** Gregory E. P. Pearcey, Lauren A. Smith, Yao Sun, E. Paul Zehr.

**Data curation:** Gregory E. P. Pearcey, Lauren A. Smith, Yao Sun.

**Formal analysis:** Gregory E. P. Pearcey, Lauren A. Smith, Yao Sun.

**Investigation:** Gregory E. P. Pearcey, Lauren A. Smith, Yao Sun.

**Methodology:** Gregory E. P. Pearcey, Lauren A. Smith, Yao Sun, E. Paul Zehr.

**Project administration:** Gregory E. P. Pearcey, Yao Sun, E. Paul Zehr.

**Resources:** E. Paul Zehr.

**Software:** Gregory E. P. Pearcey, Yao Sun, E. Paul Zehr.

**Supervision:** Gregory E. P. Pearcey, Yao Sun, E. Paul Zehr.

**Validation:** Gregory E. P. Pearcey, Yao Sun, E. Paul Zehr.

**Visualization:** Gregory E. P. Pearcey.

**Writing – original draft:** Gregory E. P. Pearcey, Lauren A. Smith.

**Writing – review & editing:** Gregory E. P. Pearcey, Yao Sun, E. Paul Zehr.

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
