## [Decision Letter · Decision Letter 0]

16 Feb 2021

PONE-D-20-38248

Even when only one limb is a player, both limbs get game: revisiting cross-education of skilled muscular control in women

PLOS ONE

Dear Dr. Zehr,

Thank you for submitting your manuscript to PLOS ONE. After careful consideration, we feel that it has merit but does not fully meet PLOS ONE’s publication criteria as it currently stands. Therefore, we invite you to submit a revised version of the manuscript that addresses the points raised during the review process.

The majority of reviewers were positive on the study while they made major suggestions to improve the clarity and on additional analyses. Please follow all in revising the manuscript.

We look forward to receiving your revised manuscript.

Kind regards,

Kei Masani

Academic Editor

PLOS ONE

Journal Requirements:

2. In your Methods section, please provide a justification for the sample size used in your study, including any relevant power calculations (if applicable).

We note that one or more of the authors are employed by a commercial company: Zanshin Consulting, Inc.

3.1. Please provide an amended Funding Statement declaring this commercial affiliation, as well as a statement regarding the Role of Funders in your study. If the funding organization did not play a role in the study design, data collection and analysis, decision to publish, or preparation of the manuscript and only provided financial support in the form of authors' salaries and/or research materials, please review your statements relating to the author contributions, and ensure you have specifically and accurately indicated the role(s) that these authors had in your study. You can update author roles in the Author Contributions section of the online submission form.

3.2. Please also provide an updated Competing Interests Statement declaring this commercial affiliation along with any other relevant declarations relating to employment, consultancy, patents, products in development, or marketed products, etc.  

Reviewers' comments:

Reviewer's Responses to Questions

**Comments to the Author**

1. Is the manuscript technically sound, and do the data support the conclusions?

Reviewer #1: No

Reviewer #2: Yes

Reviewer #3: Yes

2. Has the statistical analysis been performed appropriately and rigorously? 

Reviewer #1: No

Reviewer #2: Yes

Reviewer #3: Yes

3. Have the authors made all data underlying the findings in their manuscript fully available?

Reviewer #1: No

Reviewer #2: Yes

Reviewer #3: No

4. Is the manuscript presented in an intelligible fashion and written in standard English?

Reviewer #1: Yes

Reviewer #2: Yes

Reviewer #3: Yes

5. Review Comments to the Author

Reviewer #1: This manuscript by Pearcey et al. examines the cross-education of muscular control in terms of behavioral and neural parameters. The authors claimed both trained and untrained arms showed immense improvements in behavioral parameters, and a reduction in H-reflex.

The aim of the manuscript is interesting, but I felt the methodology was inappropriate.

Main concerns

Perez et al. (ref. 26) showed "the H-max/M-max ratio were depressed after repetition of the visuo-motor skill task and returned to baseline after 10 min". Therefore, the timing of H-reflex measurement is important for your research. It should be clearly stated in the manuscript. (I think it's better to measure the H-max/M-max ratio before and just after the game in Pre and Post -tests.)

If H-reflex was measured 10 min or more after the game, the reduction in H-reflex should be nonspecific change (i.e. not related to specific task). This is quite surprising. Why a small number of repetition of the game changed the H-reflex despite the fact that we perform skilled movements in our daily life (e.g., typing, writing, cooking, video game, playing music instruments, ...).

H-reflex was measured from the FCR. The function of the FCR is wrist flexion, wrist abduction, and forearm pronation. I guess the wrist joint was almost fixed during the game. That is, a precise activation control for the FCR are not needed during the game. I suppose the muscles whose activation timing and intensities need to be controlled precisely during the game are extensors and flexors of fingers, elbow, and shoulder. I don't think the H-reflex measurement from the FCR is appropriate for the purpose of this study.

H-max/M-max ratio was not statistically different between pre and post for the entire group (n=9). Discussion must be made on this result. The results obtained from a small number of subjects can easily change.

A "general trend for performance to improve" during three pre-tests must not be ignored. The differences in both time to completion and # of errors between Pre3 and Post were small especially for the untrained arm (Fig. 2A and 2B). If Pre4 were performed for the untrained arm, the results of Pre4 may be similar to the results actually obtained in Post judging from "a general trend for performance to improve". That is, it is not clear whether shorter time to completion and smaller number of errors obtained in Post were resulted from the cross-education effect or repetition of the test. It is a fatal problem that the present study did not provide evidence to deny this expectation.

Generally, brain activities during unimanual action using dominant hand and nondominant hand are asymmetric. Thus it is important to analyze the results from right arm training group and those from left arm training group separately.

I don't think Operation® Iron Man 2 is sufficient tool to evaluate skill for the following two reasons.

1. The buzzer sound startle the subjects, which may result in movement delay. After hearing the buzzer sound repeatedly, they will not be startled by the buzzer sound. That is, familiarity to the buzzer sound would contribute to shorten the time to completion.

2. Whether a certain amount of movement error during pulling out a plastic object buzzes or not depends on the hole because all small holes have different size and shape. The extent of movement error during every pulling out motion should be measured. Counting the buzzer sound is not enough to evaluate movement error.

Minor concerns

Abstract

L34

"the subjects" should be used instead of "women".

L35

Please explain the movement while playing the game instead of the name of the game.

L35-37

No H-max/M-max ratio reduction was found for nine subjects. （I guess）no correlation was found between H-max/M-max ratio reduction and behavioral parameters.

L39

No results gave evidence to support the claim "highlight the need to focus on female participants".

Introduction

A "generalized motor program" presented by Lashley (1942) , and the following studies are worth considering in your research.

L80

Perez et al. (ref. 26) showed "the H-max/M-max ratio were depressed after repetition of the visuo-motor skill task and returned to baseline after 10 min".

L106

If you are interested in gender effect, you should include both female and male subjects.

L122

Why did you include one left-handed subject?

L132-133

Judging from this sentence, H-reflex was measured three times (Pre1, Pre2, and Pre3). Why did you show the results of movement and neural parameters differently?（Fig. 2 and Fig. 3）

L137

No results of ECR, BB, and TB were shown.

L152

Did you use a fixed order for training session at home and in the lab? (an example of a fixed order: {lab, home, home},{lab, home, home}, ...; an example of a non-fixed order: {lab, home, home}, {home, lab, home}, ...)

L152-153

Total number of pulling out movement (55 = 11*5) should be shown clearly. It is difficult to understand the meaning of # of Errors in Fig. 2B.

L168

The effect of test on untrained arm cannot be ignored.

L254

I don't agree this interpretation of the result. A statistical test using only Pre1, Pre2, and Pre3 would show whether behavioral parameters were constant or not among three pre-tests.

L298-299

This isn't in consistent with Fig. 3F.

L330-332

No results support this statement because only female subjects participated in this study.

<improvements limb="" the="" untrained="" with="">

It is useless to compare % of skill improvements and % of strength improvements in previous studies.

<reduced excitability="" reflex="" spinal="">

The discussion is too short for the main topic of the study.

<important considerations="">

The kinematics and muscle activity during gameplay, and enough sample size are essential to the purpose of this study.</important></reduced></improvements>

Reviewer #2: This study has investigated the inter-manual transfer of motor skills using a real gaming task. The authors demonstrated that the skill learned by one hand was substantially transferred to the skill of another hand. Furthermore, the skill acquisition was likely to be accompanied by the reduction of H-reflex induced in the FCR muscle. This is a significant study showing the presence of cross-education of motor skills in a realistic situation, but I would like to raise several points that the authors need to consider for the improvement.

Major points

1) The information about the task was insufficient. The authors need to describe the details of the game and task more clearly. Which muscles and fingers did the subjects mainly use? What were the size of the objects, the holes, and the tweezers?

2) The reduction of H-reflex induced by the practice is an interesting observation. As the authors already mentioned in the manuscript, there were several limitations, like the small number of subjects in which the reflex could be measured. However, I believe that further analysis would help examine if the reduction of H-reflex was actually related to the improvement of the performance. It would be interesting to see the correlation between the amount of improvement of performance and the amount of H-reflex reduction. I am very interested in the amount of reduction of H-reflex was correlated between both hands.

Minor points

Figure 2: There must be a trade-off (speed-accuracy trade-off) between the “time to completion” and the errors. The scatter plot (e.g., x-axis is time to completion and y-axis is errors) could help the readers recognize how the practice changed the performance.

Figure 4: The current figure is not informative. Adding the recruitment curves would be helpful.

Reviewer #3: The paper by Pearcey and colleagues explores an interesting concept and is uniquely framed with reference to findings reported over 125 years ago. The phenomenon of cross-education has potential relevance to neurorehabilitation and musculoskeletal rehabilitation and the inclusion of a female-only sample addresses gaps in the current literature. The study demonstrates that unilateral training of a task with the dominant limb leads to bilateral improvement in task performance as measured by time to completion and the number of errors committed. Reductions in spinal excitability are also observed. The paper is generally well-written, but I would ask the authors to consider the following points:

1. One of the study hypotheses focuses on spinal reflex excitability, but as the Authors highlight in the introduction, transcallosal processes are likely at play. Indeed, the recent Delphi Consensus statement on cross education (Manca et al. 2021) supports the view that interhemispheric inhibition is the mechanism most likely to be mediating the phenomenon. The Authors double-down on the interhemispheric focus by introducing concepts related to stroke recovery in the present paper, where interhemispheric disinhibition is a likely contributor to motor impairment. Given the that the likely mechanisms at play are cortical in origin, it isn’t clear as to why the Authors chose to focus on measures of spinal excitability rather than cortical excitability to probe the neurophysiological underpinnings related to cross-education in muscular control. The experimental approach and hypothesis don’t quite align with putative mechanisms for the cross-education effect described by the Authors.

In lines 76-80, the use of spinal excitability assessment in the present study is justified by questioning whether there is equivalence between modulation of spinal circuitry after training for muscle control as there appears to be in muscle power. But if the primary mechanism for the phenomenon in the context of muscle power is cortically mediated, why not focus on probing whether muscular control is mediated by the same mechanism? Perhaps more justification is required in the introduction to align with the spinal reflex excitability hypothesis as postulated by the Authors.

2. By having testing and training take place using the same device, there is no evidence as to whether there is a transfer effect to other tasks. While it could be argued that ‘transfer’ is what is being examined in terms of skill development from one limb to it’s untrained contralateral homologue, the functional relevance in a clinical context would require that therapeutic benefits be transferred to other tasks. While I agree that there is potential clinical utility to this approach, I wonder whether the Authors have considered this form of transfer? Perhaps the statements related to clinical relevance should be tempered or have this point added as stipulation or limitation?

3. Were participants screened for occupational, musical or sport training that could contribute to facilitation of the learning/practice effects reported in the study?

4. Line 190 – as stated, it isn’t clear how a ‘low-level wrist flexion contraction (10% of maximum)’ was measured/monitored? Was there a load cell built into the apparatus to measure maximum and steady-state wrist flexion torque? Or was this based on EMG amplitude? Please clarify.

5. The resolution of Figure 1-3 is low. It is difficult to see what is being depicted here. Some of the axis labels and in-figure labels are illegible. Please fix this.

6. Errors and time to completion were measured during training (as stated on line 161). Were the participants aware that training performance was being monitored and were they given feedback as to their performance outcome? If so, how frequent was the feedback? To what extent could the effects observed in the present study be attributable to augmented feedback due to knowledge of results during training rather than from the cross-education effect? This consideration may warrant comment in the Discussion.

7. Figure 2A - Visual inspection of the data presented in this panel seems to indicate that time to completion for the untrained limb at Pre1, Pre2, and Pre3 is less variable and faster (generally speaking) than for the trained limb. For example, the slowest time to completion for the untrained limb at Pre1 looks to be approximately 100 seconds faster than the slowest time to completion for the trained limb. Pre2 and Pre3 seem to illustrate a similar pattern. The authors state that there is no statistically significant main effect of limb or limb x time interaction (limb x time) nor were there “significant differences between pre-test values, or between the trained and untrained limb” (Line 259). However, intuitively, one would expect that at baseline, time to completion should take longer with the untrained (non-dominant) limb. I suppose a rebuttal to this could have something to do with a speed/accuracy trade-off where participants could favour speed over accuracy. This position is somewhat supported by the Error data in panel B for Pre1; however, Pre2 and Pre3 don’t really follow. Can the authors provide an explanation for this seemingly counter-intuitive distribution of trained vs untrained outcomes at Baseline timepoints in Panels A and B?

8. The sections in the Discussion interpreting the ‘Improvements with the trained limb’ and ‘Improvements with the untrained limb’ highlight differences in the rate and extent of improvement in both limbs in the present study in comparison to what has been reported in muscle power studies (Line 337, 340, 352-356). I think the Authors could elaborate on potential mechanisms for this as the muscle control element is the apparent novel contribution here. I appreciate that the section on spinal excitability (Line 366) is meant to describe a potential mechanism, but this doesn’t really differentiate between mechanisms for muscle power vs mechanisms for muscle control.

Line 478 – Information for Reference 23 is incomplete.

Line 502 – Reference 32 – is Volume/issue information available?

6. PLOS authors have the option to publish the peer review history of their article (what does this mean?). If published, this will include your full peer review and any attached files.

Reviewer #1: No

Reviewer #2: No

Reviewer #3: No

---

## [Author Response · Author response to Decision Letter 0]

12 Jul 2021

Authors’ response: the document has been updated per the guidelines in the link above

2. In your Methods section, please provide a justification for the sample size used in your study, including any relevant power calculations (if applicable).

Authors’ response: We have added a justification of our sample size, which was based on previous investigations and reports in our lab.

We note that one or more of the authors are employed by a commercial company: Zanshin Consulting, Inc.

Authors’ response: Zanshin Consulting Inc is the company of E. Paul Zehr who is President and sole proprietor. This company purchased the games used in the research. In order to simplify we have deleted an affiliation and mention this instead in the acknowledgements. 

3.1. Please provide an amended Funding Statement declaring this commercial affiliation, as well as a statement regarding the Role of Funders in your study. If the funding organization did not play a role in the study design, data collection and analysis, decision to publish, or preparation of the manuscript and only provided financial support in the form of authors' salaries and/or research materials, please review your statements relating to the author contributions, and ensure you have specifically and accurately indicated the role(s) that these authors had in your study. You can update author roles in the Author Contributions section of the online submission form.

Authors’ response: “The funder provided support in the form of salaries for authors [insert relevant initials], but did not have any additional role in the study design, data collection and analysis, decision to publish, or preparation of the manuscript. The specific roles of these authors are articulated in the ‘author contributions’ section.”

3.2. Please also provide an updated Competing Interests Statement declaring this commercial affiliation along with any other relevant declarations relating to employment, consultancy, patents, products in development, or marketed products, etc. 

Authors’ response: This is no longer applicable. Please see above.

Authors’ response: This is no longer applicable. Please see above.

Authors’ response: We have added our data to OSF (DOI 10.17605/OSF.IO/ZFYEX – https://osf.io/zfyex/)

Authors’ response: https://orcid.org/0000-0002-1749-2924

 

Comments to the Author

Reviewer #1: 

This manuscript by Pearcey et al. examines the cross-education of muscular control in terms of behavioral and neural parameters. The authors claimed both trained and untrained arms showed immense improvements in behavioral parameters, and a reduction in H-reflex.

The aim of the manuscript is interesting, but I felt the methodology was inappropriate.

Main concerns

Perez et al. (ref. 26) showed "the H-max/M-max ratio were depressed after repetition of the visuo-motor skill task and returned to baseline after 10 min". Therefore, the timing of H-reflex measurement is important for your research. It should be clearly stated in the manuscript. (I think it's better to measure the H-max/M-max ratio before and just after the game in Pre and Post -tests.)

If H-reflex was measured 10 min or more after the game, the reduction in H-reflex should be nonspecific change (i.e. not related to specific task). This is quite surprising. Why a small number of repetition of the game changed the H-reflex despite the fact that we perform skilled movements in our daily life (e.g., typing, writing, cooking, video game, playing music instruments, ...).

Authors’ response: We appreciate the concern of the reviewer, but the work of Perez et al. was conducted to show short-term plasticity in spinal reflexes in response to a single session of visuomotor training. Indeed, in the case of Perez et al. the plasticity did not remain beyond 10 minutes, but one would not expect a single bout of training to cause long-lasting plasticity. On the contrary, in the current experiment, participants performed training multiple times per session, multiple times per week for multiple weeks. This repetitive training is what results in neuroplasticity that can persist, as is the case with operant conditioning, locomotor training, strength training, and other forms of training. In this case fine motor control was trained, and reduced spinal reflex excitability has been shown in folks who are trained for fine motor control (see Nielsen J, Crone C, Hultborn H. H-reflexes are smaller in dancers from The Royal Danish Ballet than in well-trained athletes. Eur J Appl Physiol Occup Physiol. 1993;66(2):116-21. doi: 10.1007/BF01427051. PMID: 8472692.). 

H-reflex was measured from the FCR. The function of the FCR is wrist flexion, wrist abduction, and forearm pronation. I guess the wrist joint was almost fixed during the game. That is, a precise activation control for the FCR are not needed during the game. I suppose the muscles whose activation timing and intensities need to be controlled precisely during the game are extensors and flexors of fingers, elbow, and shoulder. I don't think the H-reflex measurement from the FCR is appropriate for the purpose of this study.

Authors’ response: We thank the reviewer for their concerns, but methodological constraints limit our ability to elicit reliable H-reflexes from the distal hand muscles or muscles acting across the shoulder/elbow, and since the FCR is still involved in fine control/movement of the wrist during the game, we chose to include it as a probe for overall spinal reflex excitability. It remains for future researchers to further explore other muscles and other measures. 

H-max/M-max ratio was not statistically different between pre and post for the entire group (n=9). Discussion must be made on this result. The results obtained from a small number of subjects can easily change.

Authors’ response:

We feel the issues with H-reflex measures, including our explicit statement that effects are modest, is clearly articulated in the results and discussion. We would like to reply more effectively to the reviewer based on the last sentence but don’t understand. “Can easily change” what?

A "general trend for performance to improve" during three pre-tests must not be ignored. The differences in both time to completion and # of errors between Pre3 and Post were small especially for the untrained arm (Fig. 2A and 2B). If Pre4 were performed for the untrained arm, the results of Pre4 may be similar to the results actually obtained in Post judging from "a general trend for performance to improve". That is, it is not clear whether shorter time to completion and smaller number of errors obtained in Post were resulted from the cross-education effect or repetition of the test. It is a fatal problem that the present study did not provide evidence to deny this expectation.

Authors’ response: 

We are not completely certain we follow the reviewers main point? This is a training intervention where repeated exposure may lead to plastic adaptation and enhancement of skill. As such, it does remain possible that pre assessments were producing some skill learning. If so, then our evidence of changes due to 5 weeks of training is actually underrepresenting the true change. That is, any changes to pre make changes in post harder to find. So, we suggest it’s actually not a problem and instead makes our outcomes more conservative.

Generally, brain activities during unimanual action using dominant hand and nondominant hand are asymmetric. Thus it is important to analyze the results from right arm training group and those from left arm training group separately.

Authors’ response: We do not think it is necessary to analyze subjects based on handedness since 1) there is only one left-hand dominant subject, 2) our group showed improvement overall, and 3) all subjects followed the same trend of improvement.

I don't think Operation® Iron Man 2 is sufficient tool to evaluate skill for the following two reasons.

1. The buzzer sound startle the subjects, which may result in movement delay. After hearing the buzzer sound repeatedly, they will not be startled by the buzzer sound. That is, familiarity to the buzzer sound would contribute to shorten the time to completion.

2. Whether a certain amount of movement error during pulling out a plastic object buzzes or not depends on the hole because all small holes have different size and shape. The extent of movement error during every pulling out motion should be measured. Counting the buzzer sound is not enough to evaluate movement error.

Authors’ response: Overall, we disagree and believe that the game is sufficient to provide a proxy for motor skill learning. People learn to complete the game faster and with less error, which means it by definition is a way to evaluate skill. Also, it is a faithful reproduction of the original 1894 study where a buzzer sounded when errors were made. Nevertheless, we have added a statement to acknowledge that a reduction in startle may have contributed to decreased time to completion.

Minor concerns

Abstract

L34 "the subjects" should be used instead of "women".

Authors’ response: We disagree. The subjects are women, and therefore “women” can be used here. This is especially relevant since we are highlighting the need to study women. 

L35 Please explain the movement while playing the game instead of the name of the game.

Authors’ response: Our preference is to keep abstract details brief. We have added details about the game in the main text of the methods.

L35-37 No H-max/M-max ratio reduction was found for nine subjects. （I guess）no correlation was found between H-max/M-max ratio reduction and behavioral parameters.

Authors’ response: Hmax/Mmax ratios are a gross test of overall H-reflex excitability that is not typically sensitive to slight changes due to training. Hence, the recruitment curves are more sensitive to small changes in smaller populations of motor units within the reflex pathway.

L39 No results gave evidence to support the claim "highlight the need to focus on female participants".

Authors’ response: We disagree. The results show that cross-education occurs in a women only sample. Women-only samples are rarely studied in the cross education field, and therefore the statement stands. We are actually rather stunned to read that the reviewer does not realize how understudied women have been in this area.

Introduction

A "generalized motor program" presented by Lashley (1942) , and the following studies are worth considering in your research.

Authors’ response: Thanks for the suggestion, but we believe that there are many caveats with the notion of a generalized motor program that make it irrelevant in this case. In particular, there are issues with storage and how the CNS deals with feedback during movement that are not considered within the generalized motor program framework (see central contributions to motor control in Motor Control and Learning, eds Schmidt, Lee, Winstein, Wulf and Zelaznik).

L80

Perez et al. (ref. 26) showed "the H-max/M-max ratio were depressed after repetition of the visuo-motor skill task and returned to baseline after 10 min".

Authors’ response: We agree that this was found which is why we use the words “during acquisition of a novel fine motor skill” indicating that there is a brief period during/immediately after training where H-reflex is reduced. We have added some context.

L106

If you are interested in gender effect, you should include both female and male subjects.

Authors’ response: We were not interested in gender effects but rather the cross-education of skill in women, which we have shown. Again, the reviewer seems to not understand the base issue we are trying to address. To help with this we have added clearer language in the abstract especially.

L122

Why did you include one left-handed subject?

Authors’ response: we did not have exclusion criteria based on handedness.

L132-133

Judging from this sentence, H-reflex was measured three times (Pre1, Pre2, and Pre3). Why did you show the results of movement and neural parameters differently?（Fig. 2 and Fig. 3）

Authors’ response: We have clarified that the pre-test recruitment curve variables were normalized and combined across pre-tests (i.e. 40x3 = 120 stimuli) and compared vs post. 

L137

No results of ECR, BB, and TB were shown.

Authors’ response: It is standard practice in our lab to monitor and record activity of heteronymous muscles while recording H-reflexes to ensure remote activity does not influence our measures (please see Zehr 2002 – Considerations for use of the Hoffmann reflex in exercise studies)

L152

Did you use a fixed order for training session at home and in the lab? (an example of a fixed order: {lab, home, home},{lab, home, home}, ...; an example of a non-fixed order: {lab, home, home}, {home, lab, home}, ...)

Authors’ response: A fixed order was used, but this order differed between subjects in order to accommodate participants in the lab.

L152-153

Total number of pulling out movement (55 = 11*5) should be shown clearly. It is difficult to understand the meaning of # of Errors in Fig. 2B.

Authors’ response: We disagree. We have quantified errors based on buzzer sounds. 

L168

The effect of test on untrained arm cannot be ignored.

Authors’ response: The effects of three tests performed 5 weeks prior to the post test is unlikely to be a major contributor, rather there is likely training induced cross-education of skill.

L254

I don't agree this interpretation of the result. A statistical test using only Pre1, Pre2, and Pre3 would show whether behavioral parameters were constant or not among three pre-tests.

Authors’ response: We will have to agree to disagree with the reviewer here. This is based on our comments above and our prior (and ongoing) extensive use of multiple baseline control procedures in intervention studies. 

L298-299

This isn't in consistent with Fig. 3F.

Authors’ response: We apologize, but we don’t understand what is intended by this comment?

L330-332

No results support this statement because only female subjects participated in this study.

Authors’ response: The results support the inclusion of women-only samples since we have shown that cross-education of skill occurs in a women-only sample. We encourage others to study women-only samples as well.

It is useless to compare % of skill improvements and % of strength improvements in previous studies.

Authors’ response: We believe it is a reasonable comparison in the context it is used. In fact, relative performance assessment as a ratio or percentage is one of the most commonly used procedures to compare across studies and participants. How else would one compare strength to skill adaptation if not a relative change?

The discussion is too short for the main topic of the study.

Authors’ response: We do not agree but if the reviewer has specific recommendations on what to discuss in more detail, we are keen to listen. However, we don’t understand the suggestion to make the discussion longer because it is too short?

The kinematics and muscle activity during gameplay, and enough sample size are essential to the purpose of this study.

Authors’ response: As stated in our section on “important considerations,” we did not measure kinematics or muscle activity during gameplay, so we cannot provide this. In addition, we discuss the limitations associated with our small sample in this section, yet we have now given a rationale of our sample size, which has been added in text.

“The sample size was determined based on prior related work in the laboratory using similar designs and outcome measures, which were sufficient to achieve significant cross-education effects of strength training with moderate effect sizes (Barss et al 2018 doi: 10.1152/japplphysiol.00390.2017, Sun et al 2018 doi: 10.1007/s00221-018-5275-6).”

 

Reviewer #2: 

This study has investigated the inter-manual transfer of motor skills using a real gaming task. The authors demonstrated that the skill learned by one hand was substantially transferred to the skill of another hand. Furthermore, the skill acquisition was likely to be accompanied by the reduction of H-reflex induced in the FCR muscle. This is a significant study showing the presence of cross-education of motor skills in a realistic situation, but I would like to raise several points that the authors need to consider for the improvement.

Authors’ response: Thanks for the constructive feedback, which we believe helped improve the quality of the manuscript.

Major points

1) The information about the task was insufficient. The authors need to describe the details of the game and task more clearly. Which muscles and fingers did the subjects mainly use? What were the size of the objects, the holes, and the tweezers?

Authors’ response: We have added some more details about the game. In particular, we briefly described that participants were not instructed how to hold the tweezers, but the most common way was to hold the tweezers between the tips of the thumb and index finger with support from the middle finger. Objects ranged in size with graspable regions ranging from 1-4mm. The tweezers were 5.4cmm in length with a 7mm aperture. The holes ranged in size, but the narrowest region was 16mm.

2) The reduction of H-reflex induced by the practice is an interesting observation. As the authors already mentioned in the manuscript, there were several limitations, like the small number of subjects in which the reflex could be measured. However, I believe that further analysis would help examine if the reduction of H-reflex was actually related to the improvement of the performance. It would be interesting to see the correlation between the amount of improvement of performance and the amount of H-reflex reduction. I am very interested in the amount of reduction of H-reflex was correlated between both hands.

Authors’ response: Although these aspects are interesting to explore, the small sample size severely limits our ability to run correlational/regression analyses. In addition, we have no metric of overall improvement of performance, which complicates an answer these questions (i.e. improvement in speed vs errors). We think that the change in the reflex excitability is a minor side story and have tried to downplay this finding, particularly in response to another reviewers concerns.

Minor points

Figure 2: There must be a trade-off (speed-accuracy trade-off) between the “time to completion” and the errors. The scatter plot (e.g., x-axis is time to completion and y-axis is errors) could help the readers recognize how the practice changed the performance.

Authors’ response: We are unsure what the reviewer thinks the scatter below adds to the story. Is this what is suggested?

Figure 4: The current figure is not informative. Adding the recruitment curves would be helpful.

Authors’ response: We do not agree with the reviewer and prefer Figure 4 as it is. It currently shows the relative amplitudes and shapes of waveforms (i.e. M-wave and H-reflexes) along with the latencies of the responses. True raw data is beneficial to ensure replicability in the future.

 

Reviewer #3: 

The paper by Pearcey and colleagues explores an interesting concept and is uniquely framed with reference to findings reported over 125 years ago. The phenomenon of cross-education has potential relevance to neurorehabilitation and musculoskeletal rehabilitation and the inclusion of a female-only sample addresses gaps in the current literature. The study demonstrates that unilateral training of a task with the dominant limb leads to bilateral improvement in task performance as measured by time to completion and the number of errors committed. Reductions in spinal excitability are also observed. The paper is generally well-written, but I would ask the authors to consider the following points:

Authors’ response: Thanks for the helpful suggestions, which we think help improved the quality of the manuscript.

1. One of the study hypotheses focuses on spinal reflex excitability, but as the Authors highlight in the introduction, transcallosal processes are likely at play. Indeed, the recent Delphi Consensus statement on cross education (Manca et al. 2021) supports the view that interhemispheric inhibition is the mechanism most likely to be mediating the phenomenon. The Authors double-down on the interhemispheric focus by introducing concepts related to stroke recovery in the present paper, where interhemispheric disinhibition is a likely contributor to motor impairment. Given the that the likely mechanisms at play are cortical in origin, it isn’t clear as to why the Authors chose to focus on measures of spinal excitability rather than cortical excitability to probe the neurophysiological underpinnings related to cross-education in muscular control. The experimental approach and hypothesis don’t quite align with putative mechanisms for the cross-education effect described by the Authors.

Authors’ response: We certainly agree that cortical mechanisms are major factors mediating cross-education. However, cortical output must arrive at and pass through the spinal cord to be expressed as motor activity. Thus, spinal adaptations are also relevant to study. We also suggest that our study was not intended as a major mechanistic investigation.

In lines 76-80, the use of spinal excitability assessment in the present study is justified by questioning whether there is equivalence between modulation of spinal circuitry after training for muscle control as there appears to be in muscle power. But if the primary mechanism for the phenomenon in the context of muscle power is cortically mediated, why not focus on probing whether muscular control is mediated by the same mechanism? Perhaps more justification is required in the introduction to align with the spinal reflex excitability hypothesis as postulated by the Authors.

Authors’ response: The measurement of spinal excitability was meant to be an additional probe to examine whether this type of training influences neural circuits within the spinal cord. The main message remains that cross-education of skill occurs in women. A change in spinal excitability should not take away from this message, but rather provide some additional insights about what might contribute to this finding. We have made an attempt to de-emphasize the importance of the reflex findings.

2. By having testing and training take place using the same device, there is no evidence as to whether there is a transfer effect to other tasks. While it could be argued that ‘transfer’ is what is being examined in terms of skill development from one limb to it’s untrained contralateral homologue, the functional relevance in a clinical context would require that therapeutic benefits be transferred to other tasks. While I agree that there is potential clinical utility to this approach, I wonder whether the Authors have considered this form of transfer? Perhaps the statements related to clinical relevance should be tempered or have this point added as stipulation or limitation?

Authors’ response: This is an excellent point that we did consider addressing with a separate game. Unfortunately, only a small subset (n = 2) of subjects completed the “transfer” game at both time points (pre/post), which was too small of a sample to draw conclusions about the transfer. We have made an attempt to temper our conclusions about the clinical relevance of the cross-education of skill based on this game.

3. Were participants screened for occupational, musical or sport training that could contribute to facilitation of the learning/practice effects reported in the study?

Authors’ response: we used a convenience sample from the undergraduate population of the University of Victoria and participants were not screened based on occupational, musical or sport training. 

4. Line 190 – as stated, it isn’t clear how a ‘low-level wrist flexion contraction (10% of maximum)’ was measured/monitored? Was there a load cell built into the apparatus to measure maximum and steady-state wrist flexion torque? Or was this based on EMG amplitude? Please clarify.

Authors’ response: Thanks for pointing out this omission of details. We have added that there was EMG feedback given to participants in the following way: “Feedback consisted of a 100 ms moving average of the rectified EMG.”

5. The resolution of Figure 1-3 is low. It is difficult to see what is being depicted here. Some of the axis labels and in-figure labels are illegible. Please fix this.

Authors’ response: We are not sure how this occurred, because on our end, the figures are high-resolution and very clear. We hope that the figures are clear on this second submission.

6. Errors and time to completion were measured during training (as stated on line 161). Were the participants aware that training performance was being monitored and were they given feedback as to their performance outcome? If so, how frequent was the feedback? To what extent could the effects observed in the present study be attributable to augmented feedback due to knowledge of results during training rather than from the cross-education effect? This consideration may warrant comment in the Discussion.

Authors’ response: As stated “Errors and completion time were only recorded in lab training sessions.” No feedback was provided to participants about the number of errors that they committed. In this way, participants could only understand the number of errors if they counted them in their own heads, which was not apparent based on the observations of our team of investigators. We have clarified this in the methods, but we do not believe that we should add anything on this point into the discussion.

7. Figure 2A - Visual inspection of the data presented in this panel seems to indicate that time to completion for the untrained limb at Pre1, Pre2, and Pre3 is less variable and faster (generally speaking) than for the trained limb. For example, the slowest time to completion for the untrained limb at Pre1 looks to be approximately 100 seconds faster than the slowest time to completion for the trained limb. Pre2 and Pre3 seem to illustrate a similar pattern. The authors state that there is no statistically significant main effect of limb or limb x time interaction (limb x time) nor were there “significant differences between pre-test values, or between the trained and untrained limb” (Line 259). However, intuitively, one would expect that at baseline, time to completion should take longer with the untrained (non-dominant) limb. I suppose a rebuttal to this could have something to do with a speed/accuracy trade-off where participants could favour speed over accuracy. This position is somewhat supported by the Error data in panel B for Pre1; however, Pre2 and Pre3 don’t really follow. Can the authors provide an explanation for this seemingly counter-intuitive distribution of trained vs untrained outcomes at Baseline timepoints in Panels A and B?

Authors’ response: This is a great observation, and at first glance is quite puzzling. Indeed, it would be strange if the non-dominant limb was better at gameplay than the dominant limb. To understand this seemingly strange observation, one must consider the order of task completion. In all cases, the participants played the game with the dominant (trained) limb before playing with the non-dominant (untrained) limb. We believe that this order of gameplay most likely explains the visual trend in the data for the non-dominant limb to be faster (albeit with more errors in Pre1) than the dominant limb.

8. The sections in the Discussion interpreting the ‘Improvements with the trained limb’ and ‘Improvements with the untrained limb’ highlight differences in the rate and extent of improvement in both limbs in the present study in comparison to what has been reported in muscle power studies (Line 337, 340, 352-356). I think the Authors could elaborate on potential mechanisms for this as the muscle control element is the apparent novel contribution here. I appreciate that the section on spinal excitability (Line 366) is meant to describe a potential mechanism, but this doesn’t really differentiate between mechanisms for muscle power vs mechanisms for muscle control.

Authors’ response: It is our belief that strength is a skill, too. However training for strength and training for muscular control differ in the skill that is being performed. With strength, plasticity occurs to increase the force generating capacity of a particular group of muscles in a particular direction. This can involve facilitation of agonist and synergist muscles, while inhibiting antagonists. On the other hand, training for muscular control essentially involves training inhibitory circuits to ensure fine control and smooth movements, while enhancing the specific motor plan for the skilled movement that is trained. We have expanded our discussion to point out these subtleties.

Line 478 – Information for Reference 23 is incomplete.

Authors’ response: Our apologies – this must have been an issue with our reference manager. The reference should be:

Manca A, Hortobágyi T, Carroll TJ, Enoka RM, Farthing JP, Gandevia SC, Kidgell DJ, Taylor JL, Deriu F. Contralateral Effects of Unilateral Strength and Skill Training: Modified Delphi Consensus to Establish Key Aspects of Cross-Education. Sports Med. 2021 Jan;51(1):11-20. doi: 10.1007/s40279-020-01377-7. PMID: 33175329; PMCID: PMC7806569.

Line 502 – Reference 32 – is Volume/issue information available?

Authors’ response: Our apologies, again – this must have been another issue with our reference manager. The reference should be:

Sun Y, Ledwell NMH, Boyd LA, Zehr EP. Unilateral wrist extension training after stroke improves strength and neural plasticity in both arms. Exp Brain Res. 2018 Jul;236(7):2009-2021. doi: 10.1007/s00221-018-5275-6. Epub 2018 May 5. PMID: 29730752.

---

## [Decision Letter · Decision Letter 1]

8 Sep 2021

PONE-D-20-38248R1

1894 revisited: Cross-education of skilled muscular control in women and the importance of representation

PLOS ONE

Dear Dr. Zehr,

Thank you for submitting your manuscript to PLOS ONE. After careful consideration, we feel that it has merit but does not fully meet PLOS ONE’s publication criteria as it currently stands. Therefore, we invite you to submit a revised version of the manuscript that addresses the points raised during the review process.

Reviewer 1 believes that FCR possibly does not play a role in game task, and requests evidence showing the role of FCR in the task provided to the participants. Furthermore, reviewer 1 has also critiqued the overall experimental design, which the Academic Editor does not support too. Please provide clear justifications and/or describe them as the study limitations in the manuscript as well as in the response letter to the reviewer. Also, please respond to the minor comments of reviewer 3

We look forward to receiving your revised manuscript.

Kind regards,

Lucinda Shen 

Staff Editor

On behalf of 

Kei Masani

Academic Editor

PLOS ONE

Journal Requirements:

Reviewers' comments:

Reviewer's Responses to Questions

**Comments to the Author**

1. If the authors have adequately addressed your comments raised in a previous round of review and you feel that this manuscript is now acceptable for publication, you may indicate that here to bypass the “Comments to the Author” section, enter your conflict of interest statement in the “Confidential to Editor” section, and submit your "Accept" recommendation.

Reviewer #1: (No Response)

Reviewer #2: All comments have been addressed

Reviewer #3: (No Response)

2. Is the manuscript technically sound, and do the data support the conclusions?

Reviewer #1: No

Reviewer #2: Yes

Reviewer #3: Yes

3. Has the statistical analysis been performed appropriately and rigorously? 

Reviewer #1: Yes

Reviewer #2: Yes

Reviewer #3: Yes

4. Have the authors made all data underlying the findings in their manuscript fully available?

Reviewer #1: Yes

Reviewer #2: Yes

Reviewer #3: Yes

5. Is the manuscript presented in an intelligible fashion and written in standard English?

Reviewer #1: Yes

Reviewer #2: Yes

Reviewer #3: Yes

6. Review Comments to the Author

Reviewer #1: The authors failed to provide a rationale for associating the results of H-reflex in FCR to behavioral results. The authors did not provide any information about the role FCR plays in the gameplay. My expectation is that in the gameplay the wrist movement is small and rather fixed. That is, the wrist muscles including FCR would not contribute much to skill acquisition/ fine motor control.

Both the time to completion and the number of errors continued to decrease (improve) in the three pre assessments (Fig. 2AB). In other words, repeated assessment improves behavioral performance. The fourth assessment was conducted after the training period, but it is possible that the same results would have been obtained if it had been conducted before the training period. It is a fatal problem that the three pre assessments did not provide reliable baseline.

Reviewer #2: The authors have addressed my concerns.

I also thank the authors for making the scatter plot showing the relationship between the completion time and the number of errors. The authors do not need to include this plot because I understood the information is not so different from Figure 3.

Reviewer #3: The Authors have generally addressed my concerns, but I do make the following comments for consideration:

1. Regarding my previous comment about the counter-intuitive observations of the data presented in Figure 2A, it should be clarified in the Methods that participants always performed the muscular control assessments with the Trained hand first. As an aside, I wonder whether the Authors’ response and acknowledgement of Reviewer 1’s comment about startle in the Limitations section may also contribute to my point regarding Figure 2A. Performing the task with the Trained hand first and experiencing errors (i.e. buzzers sounds) may increase time to completion initially, but may then lead to an attenuation of the effect by the time participants performed the task with the Untrained hand.

2. I note that References 32 and 33 are duplicates. Please correct.

3. The resolution of the images is still poor, but I this could just be on my end.

7. PLOS authors have the option to publish the peer review history of their article (what does this mean?). If published, this will include your full peer review and any attached files.

Reviewer #1: No

Reviewer #2: No

Reviewer #3: No

---

## [Author Response · Author response to Decision Letter 1]

10 Dec 2021

Decision point #1: Reviewer 1 believes that FCR possibly does not play a role in game task, and requests evidence showing the role of FCR in the task provided to the participants. 

Comment from reviewer 1: The authors failed to provide a rationale for associating the results of H-reflex in FCR to behavioral results. The authors did not provide any information about the role FCR plays in the gameplay. My expectation is that in the gameplay the wrist movement is small and rather fixed. That is, the wrist muscles including FCR would not contribute much to skill acquisition/ fine motor control.

Authors’ response: Although the direct role of FCR in gameplay may be limited, its role in stabilization and control of wrist movements is relevant to improving the number of errors. For instance, the FCR works in synergy with the ECR to produce radial deviation, and the FCR acts to stabilize and flex the wrist. To better understand the role of FCR in gameplay, one should consider gain control of the limb during a task being trained. If gain is high, small adjustments in neural drive would result in large adjustments in kinematics, potentially resulting in errors. On the contrary, if training results in reducing the gain of muscles acting at the wrist (e.g. the FCR muscle), adjustments in synaptic input (descending neural drive or sensory input as examples) would simply result in smaller deviations in the kinematics and, thus, reduced errors. Therefore, although not the primary muscle used during gameplay to make fine motor adjustments, the FCR is an appropriate muscle to act as a proxy for gain to arm muscles distal to the elbow. If the reviewer disagrees, we would be happy to revise the manuscript with their suggestion in mind.

In any case we have added some rationale for the choice of this muscle (lines 180-186) and emphasized this as a limitation to our study, which can be found in the “important considerations” section of the Discussion (lines 441-447).

Decision point #2: Reviewer 1 has also critiqued the overall experimental design, which the Academic Editor does not support too. Please provide clear justifications and/or describe them as the study limitations in the manuscript as well as in the response letter to the reviewer.

Comment from reviewer 1: Both the time to completion and the number of errors continued to decrease (improve) in the three pre assessments (Fig. 2AB). In other words, repeated assessment improves behavioral performance. The fourth assessment was conducted after the training period, but it is possible that the same results would have been obtained if it had been conducted before the training period. It is a fatal problem that the three pre assessments did not provide reliable baseline.

Authors’ response: This is an interesting point but we are also slightly confused. The reviewer seems to be arguing that the training exposures didn’t actually train the participants and that without the training a 4th assessment would have yielded a training effect on its own. Is this correct? 

Also, please note that when using statistical tests things are either significantly different or they are not. There were not significant changes across baseline tests according to statistical evaluations. Therefore the comment that measures “continued to decrease (improve)” is disingenuous. Even if this were true, and statistical testing supported the reviewer’s suggestion of perceived differences, this actually means we have underestimated the power of the training intervention and are actually being more conservative in our assessments. If the reviewer is asking us to say that our non-significant differences across baseline are “significant”, and we extend this logic, would we then be asked to do the same for any of our experimental measures?

In any case, we think we understand what the reviewer is getting at and have added “Although there were no significant changes across baseline measures, visual inspection of our data may suggest a trend towards improvement in these sessions. Future studies may consider multiple post-test measures as a way to evaluate this.” to the “important considerations” section (lines 422-425).

Decision point #3: Please respond to the minor comments of reviewer 3

1. Regarding my previous comment about the counter-intuitive observations of the data presented in Figure 2A, it should be clarified in the Methods that participants always performed the muscular control assessments with the Trained hand first. As an aside, I wonder whether the Authors’ response and acknowledgement of Reviewer 1’s comment about startle in the Limitations section may also contribute to my point regarding Figure 2A. Performing the task with the Trained hand first and experiencing errors (i.e. buzzers sounds) may increase time to completion initially, but may then lead to an attenuation of the effect by the time participants performed the task with the Untrained hand.

Authors’ response: We agree with both points, and as such have added a point in the methods (line 167) about the test order, and in the further considerations (lines 432-434) about how the habituation of the startle resulting from the buzzer may have contributed to the counter-intuitive results of the non-dominant hand completing the game at a faster rate.

2. I note that References 32 and 33 are duplicates. Please correct.

Authors’ response: We have revised by removing 33.

3. The resolution of the images is still poor, but I this could just be on my end.

Authors’ response: We apologize for this. The uploaded figures are clear on our end. We will ensure they are clear if the manuscript is accepted. We have uploaded .tif files exported at 300DPI from Adobe Illustrator, for the record.

---

## [Editor Report · Decision Letter 2]

16 Feb 2022

1894 revisited: Cross-education of skilled muscular control in women and the importance of representation

PONE-D-20-38248R2

Dear Dr. Zehr,

We’re pleased to inform you that your manuscript has been judged scientifically suitable for publication and will be formally accepted for publication once it meets all outstanding technical requirements.

Kind regards,

Kei Masani

Academic Editor

PLOS ONE
---

## [Editor Report · Acceptance letter]

8 Mar 2022

PONE-D-20-38248R2 

1894 revisited: Cross-education of skilled muscular control in women and the importance of representation 

Dear Dr. Zehr:

I'm pleased to inform you that your manuscript has been deemed suitable for publication in PLOS ONE. Congratulations! Your manuscript is now with our production department. 

Kind regards, 

on behalf of

Dr. Kei Masani 

Academic Editor

PLOS ONE